# DC-PINNs:
# Physics-Informed Neural Networks for Solving Derivative-Constrained PDEs

## Abstract

Physics-Informed Neural Networks (PINNs) have emerged as a promising approach for solving partial differential equations (PDEs) using deep learning. However, standard PINNs do not address the problem of constrained PDEs, where the solution must satisfy additional equality or inequality constraints beyond the governing equations. In this paper, we introduce Derivative-Constrained PINNs (DC-PINNs), a novel framework that seamlessly incorporates constraint information into the PINNs training process. DC-PINNs employ a constraint-aware loss function that penalizes constraint violations while simultaneously minimizing the PDE residual. Key components include self-adaptive loss balancing techniques that automatically tune the relative weighting of each term, enhancing training stability, and the use of automatic differentiation to efficiently compute derivatives. This study demonstrates the effectiveness of DC-PINNs on several benchmark problems, from basic to complex, such as quantitative finance and applied physics, including heat diffusion, volatility surface calibration, and incompressible flow dynamics. The results showcase improvements in generating solutions that satisfy the constraints compared to baseline PINNs methods. The DC-PINNs framework opens up new possibilities for solving constrained PDEs in multi-objective optimization problems.

## 1 Introduction

Partial differential equations (PDEs) play a crucial role in modelling various physical phenomena across scientific and engineering disciplines. Traditional numerical methods for solving PDEs, such as finite difference and finite element methods, have been widely used but often face challenges in terms of computational efficiency and handling complex geometries. Recently, Physics-Informed Neural Networks (PINNs) by Raissi et al. (2019) have emerged as a promising alternative, leveraging deep learning to solve PDEs with high accuracy and efficiency.

However, despite their empirical successes, PINNs still face challenges in the presence of additional equality or inequality constraints beyond the PDE itself, as pioneered in Lagaris et al. (1998). Such constrained PDEs are ubiquitous in real-world applications: heat diffusion with temperature bounds, option pricing with no-arbitrage constraints, and fluid flow with velocity limits, to name a few. Naive techniques for embedding constraints can lead to unstable training, slow convergence, and constraint violation. Although more sophisticated approaches have been proposed to better handle constraints in PINNs, Conservative PINNs Jagtap et al. (2020) and DC NN Lo & Huang (2023) enforce equality constraints, while Augmented Lagrangian approaches Lu et al. (2021) and theory-guided neural networks Chen et al. (2021) show stronger performance on inequality-constrained PDEs. Nevertheless, there is still a need for methods that can effectively handle constraints while reducing the dependence on intricate hyperparameter adjustments and problem-specific architectures.

In this study, we propose Derivative-Constrained PINNs (DC-PINNs), a general and robust framework for solving derivative-constrained PDEs using deep learning. Our approach seamlessly integrates the constraints into the learning process, ensuring that the solution satisfies the prescribed conditions while maintaining the benefits of PINNs. The key contributions of this study include:
- A flexible constraint-aware loss function that admits general nonlinear constraints and seamlessly incorporates them into the PDE residual objective.

- Self-adaptive loss balancing techniques that automatically tune the weightings of the objective terms, including derivatives obtained through automatic differentiation stabilizing training across diverse problem settings.
- Demonstration of the approximation ability of DC-PINNs on benchmark PDEs from simple to complex, including the heat equations, structural modelling of finance, and fluid dynamics problem, showcasing improvements over PINNs approaches.

The study aims to highlight the importance of explicit constraint handling in PINNs, as relying solely on the PDE residual term may lead to solutions that violate critical physical principles. The extended approach can explore applicable to several key areas: 1. Basic PDE problems with added physical insights  2. Problems requiring multiple derivative constraints to avoid system failure  3. Complex problems with diverse internal dynamics that typically resist convergence. This study provides experiments for these purposes on heat equations, local volatility models in finance, and incompressible flows in fluid dynamics. Comparisons with existing techniques show our approach manages constraints and maintains physical consistency more effectively.

These experiments compare our framework against existing approaches in handling complex constraints and maintaining physical consistency. Recasting derivative-constrained PDE solving as a multi-objective optimisation problem opens up new possibilities for solving a wider range of physically constrained systems with improved accuracy and reliability.

## 2 PROBLEM FORMULATION

### 2.1 DERIVATIVE-CONSTRAINED PDE PROBLEM

Physics-based machine learning can be formulated as an optimization problem that aims to find the feasible parameters $\hat{\theta}$ of a parameterized function $u := \varphi_\theta(x)$ while satisfying constraints represented by PDEs and related conditions by

$$\hat{\theta} = \underset{\theta}{\operatorname{argmin}} \ \mathcal{L}(x, \mathcal{D}y), \tag{1}$$

$$\text{s.t.} \quad f(x, \mathcal{D}y) = 0, \quad \text{governing PDEs}$$

$$\mathcal{B} = \left\{ b_k, \mathcal{D}_k^{(b)} | \mathcal{D}_k^{(b)} y(x) = 0, x \in b_k \right\}_{k=1}^{N_b} \tag{2}$$

$$\mathcal{H} = \left\{ h_k, \mathcal{D}_k^{(h)} | \mathcal{D}_k^{(h)} y(x) \geq 0, x \in h_k \right\}_{k=1}^{N_h}$$

where $\mathcal{L}$ is the objective loss function, multivariate inputs $x \in \mathbb{R}^n$ are state variables, $\mathcal{D}$ is the set of individual differential operators including the 0-th order, $f$ represents the governing PDEs, $\mathcal{B}$ is the set of boundary equality constraints, $\mathcal{H}$ is the set of inequality constraints, and $N_{(\cdot)}$ denotes the number of corresponding conditions.

This study focuses on inequality constraints involving derivatives and employs a discretize-then-optimize approach combined with gradient-based optimisation using artificial neural networks (ANNs). The optimization problem is first discretized numerically, transforming it into a finite-dimensional problem. To ensure that the smooth ANN solution complies with the governing equations and constraints for PDEs involving derivatives of input variables, this study uses the multilayer perceptron (MLP) with Automatic Differentiation for gradient calculation, requiring at least second-order differentiable activation functions. For PDEs involving $\dot{n}$-th order partial derivatives, the entire MLP must be $(\dot{n}+1)$-th order differentiable to facilitate gradient-based optimization effectively.

### 2.2 PHYSICS-INFORMED NEURAL NETWORKS (PINNS)

Physics-Informed Neural Networks (PINNs), introduced in Raissi et al. (2019), are neural networks that incorporate underlying physical laws into the architecture through PDEs, forming a new class of data-efficient universal function approximations. We consider a parametrized PDE system given by

$$\boldsymbol{f}[\varphi(x)], x \in \Omega,$$
$$\boldsymbol{b}[\varphi(x)], x \in \partial\Omega, \tag{3}$$

where $\boldsymbol{f}$ and $\boldsymbol{b}$ are a set of PDE and boundary operators, $\Omega$ and $\partial\Omega$ are the spatial domain and the boundary.

PINNs solve this PDE system as an optimization problem using an artificial neural network by minimizing the total loss in a deep learning context:

$$\mathcal{L} := \mathcal{L}_0 + \mathcal{L}_b + \mathcal{L}_f, \tag{4}$$

$$\mathcal{L}_0 := \frac{1}{N_0} \sum_{i=1}^{N_0} \left| \varphi(x_0^{(i)}) - \hat{y}_0^{(i)} \right|^2, \mathcal{L}_b := \frac{1}{N_b} \sum_{i=1}^{N_b} \left| b[\varphi(x_b^{(i)})] \right|^2, \mathcal{L}_f := \frac{1}{N_f} \sum_{i=1}^{N_f} \left| f[\varphi(x_f^{(i)})] \right|^2, \tag{5}$$

where $\mathcal{L}_0$ represents the error between observed ($\hat{y}$) and predicted values, $\mathcal{L}_b$ enforces boundary conditions, and $\mathcal{L}_f$ penalizes the PDE residual at a set of collocation points.

## 2.3 DERIVATIVE-CONSTRAINED PINNS (DC-PINNS)

We now consider the extension of PINNs to handle derivative inequality constraints, which we term Derivative-Constrained PINNs (DC-PINNs). Assume the presence of inequality constraints of the form

$$\boldsymbol{h}[\varphi(x)], x \in \Omega \tag{6}$$

where $\boldsymbol{h}[\cdot]$ represents a set of differential operators acting on an inequality equation, which includes derivatives. There are several methods available to enforce inequality conditions in general. The direct approach is to formulate loss functions of inequalities and impose them as soft constraints with fixed loss weights. To fit with inequality constraints, a loss $\mathcal{L}_h$ to be minimized is defined as

$$\mathcal{L}_h := \frac{1}{N_h} \sum_{i=1}^{N_h} \gamma \circ \left| h\left(\varphi(x_h^{(i)})\right) \right|^2. \tag{7}$$

$$\gamma(x) = \begin{cases} x, & \text{if inequality is not satisfied} \\ 0, & \text{otherwise} \end{cases} \tag{8}$$

where $\mathcal{L}_h$ penalizes the violation of inequality constraints at a set of collocation points, and the function $\gamma$ determines the penalty based on whether the inequality is satisfied or not.

Our proposed framework extends PINNs to handle derivative-constrained PDEs effectively by seamlessly integrating the constraints into the learning process. We introduce a constraint-aware loss function that penalizes violations of the constraints while simultaneously minimizing the residual loss. However, setting large loss weights can cause an ill-conditioned problem. On the other hand, when small loss weights are chosen, the estimated solution may violate the inequalities. In this sense, we formulate the total cost in DC-PINNs to be minimized as,

$$\mathcal{L} := \lambda_0 \hat{\mathcal{L}}_0 + \lambda_b \hat{\mathcal{L}}_b + \lambda_f \hat{\mathcal{L}}_f + \lambda_h \hat{\mathcal{L}}_h, \tag{9}$$

where $\lambda$ are weighting coefficients for each categorized loss term. By minimizing this total loss, the neural network approximation satisfies the governing PDE, boundary/initial conditions, and the prescribed inequality constraints. The categorized loss terms are defined as

$$\hat{\mathcal{L}}_0 = \frac{1}{N_0} \sum_{i=1}^{N_0} m_0^{(i)} \left| \varphi_\theta\left(x_0^{(i)}\right) - y_0^{(i)} \right|^2, \hat{\mathcal{L}}_b = \frac{1}{N_b} \sum_{i=1}^{N_b} m_b^{(i)} \left| \varphi_\theta\left(x_b^{(i)}\right) - y_b^{(i)} \right|^2, \tag{10}$$

$$\hat{\mathcal{L}}_f = \frac{1}{N_f} \sum_{i=1}^{N_f} m_f^{(i)} \left| f\left(\varphi_\theta(x_f^{(i)})\right) \right|^2, \hat{\mathcal{L}}_h := \frac{1}{N_h} \sum_{i=1}^{N_h} m_h^{(i)} \left| \gamma \circ h\left(\varphi_\theta(x_h^{(i)})\right) \right|^2, \tag{11}$$

where $y_0$ is the observed values, $i = 1, \ldots, N_0$ from the observed dataset, and $\hat{\mathcal{L}}_h$ represents a penalty term corresponding to inequality constraints stated the third term in equation 2. The modifications from loss configurations of standard PINNs are the introducing multipliers $\lambda$ for each categorized loss as loss terms and the weights $m$ for each individual loss for the outputs on each state variable in the categorized loss.

## 3 MULTI-OBJECTIVE OPTIMIZATION

The choice of weight coefficients for the different loss components requires careful tuning to balance the contributions of the residual loss, boundary/initial condition loss, and constraint-aware loss. As mentioned in Lu et al. (2021), if the multiplier for the categorized loss is larger, the constraint violations are penalized more severely, forcing the solutions to better satisfy the constraints. However, when the penalty coefficients are too large, the optimization problem becomes ill-conditioned and difficult to converge to a minimum. On the other hand, if the penalty coefficients are too small, then the obtained solution will not satisfy the constraints and thus is not a valid solution. Although the soft-constraint approach has worked well for inverse problems to match observed measurements, it cannot be used in general because we cannot determine appropriate multipliers in the learning process.

To enhance the usability and robustness of the framework, automated techniques for optimal weight selection are employed. This study, involves a combination of two balanced processes in the learning process, inspired by McClenny & Braga-Neto (2020); Wang et al. (2023). The first balancing intensifies the gradient of individual losses in categorized losses to enhance local constraints, especially the objective for inequality derivative constraints. The second balancing addresses the multi-scale imbalance between categorized losses in (10~11), particularly to mitigate the changing of gradient values from epoch to epoch due to the inequality feature in equation 11.

### 3.1 INDIVIDUAL LOSS BALANCING

In alignment with the neural network philosophy of self-adaptation, this study applies a straightforward procedure with fully trainable weights to generate multiplicative soft-weighting and attention mechanisms. The first balancing proposes self-adaptive weighting that updates the loss function weights via gradient ascent concurrently with the network weights. We minimize the total cost with respect to $\theta$ but also maximize it with respect to the self-adaptation weight vectors $m$ at the $k$-th epoch,

$$m_\beta^{(j)}(k+1) = m_\beta^{(j)}(k) + \eta_m \nabla_{m_\beta^{(j)}} \hat{\mathcal{L}}_\beta(k).$$ (12)

where $\beta$ specifies each loss $\beta \in \{0, b, f, h\}$ and $\eta_m$ is learning parameter. In the learning step, the derivatives with respect to self-adaptation weights are increased when the constraints are violated and become larger when the errors are larger.

### 3.2 CATEGORIZED LOSS BALANCING

Parallely, for the second balancing, loss-balancing employs the following weighting function with balancing parameters $\lambda$ in the loss function based on Eq equation 9. Considering updated at the $k$-th epoch,

$$\lambda_\beta(k+1) = \begin{cases} 1, & \text{if } \overline{\left|\nabla_\theta \hat{\mathcal{L}}_\beta(k)\right|} = 0 \\ \lambda_\beta(k) + \frac{\sum_\beta \overline{\left|\nabla_\theta \hat{\mathcal{L}}_\beta(k)\right|}}{\overline{\left|\nabla_\theta \hat{\mathcal{L}}_\beta(k)\right|}}, & \text{otherwise} \end{cases}$$ (13)

where $\nabla_i$ is the partial derivative vector (gradient) with respect to the $i$-th input vector (or value), and $\overline{|\cdot|}$ indicates the average of the absolute values of the elements in the vector. Note that the main reason for the choice of absolute values, instead of squared values as in Wang et al. (2023), is to avoid overlooking outlier violations of the inequality constraints because most elements of $\nabla \mathcal{L}_h$ are assumed to be zero values in almost all cases.

The applied methods automatically adjust the weights of loss terms based on their relative magnitudes during training at user-specified intervals. These adaptive approaches have the potential to ensure a balanced contribution of each term of unstable inequality losses to the optimization process, potentially improving convergence and accuracy.

## 4 ALGORITHMS

This section introduces the algorithm of the DC-PINNs for multi-objective problems, which control the inequity loss of the partial derivatives of a neural network function with respect to its input

features and apply the combination of loss balancing techniques for both categorized and individual losses.

---

**Algorithm 1** DC-PINNs with Balancing Processes

---

**Input:** Dataset $(x_{0,b}, y_{0,b})$, $x_f$, $x_h$, $\eta$, $\eta_m$, $p_m$, $p_\lambda$, $k_{\max}$

Consider a deep NN $\varphi_\theta(x)$ with $\theta$, and a loss function

$$\mathcal{L} := \sum_\beta \lambda_\beta \hat{\mathcal{L}}_\beta \left( m_\beta, x_\beta \left(, y_\beta \right) \right),$$

where $\hat{\mathcal{L}}_\beta$ denotes the categorized loss with $\beta \in \{0, b, f, h\}$, $m_\beta = \mathbf{1}$ are soft-weighting vectors for individual losses and $\lambda_\beta = 1$ are dynamic multipliers.

**for** $k = 1, \ldots, k_{\max}$ **do**

    Compute $\nabla_\theta \hat{\mathcal{L}}_\beta(k)$ by automatic differentiation

    **if** $k \equiv 0 \mod p_m$ **then**

        Update $m_\beta$ by

$$m_\beta^{(j)}(k+1) = m_\beta^{(j)}(k) + \eta_m \nabla_{m_\beta^{(j)}} \hat{\mathcal{L}}_\beta(k),$$

        where $m_\beta(k), \hat{\mathcal{L}}_\beta(k)$ shows values at $k$-th iteration.

    **end if**

    **if** $k \equiv 0 \mod p_\lambda$ **then**

        Update $\lambda_\beta$ by

$$\lambda_\beta(k+1) = \begin{cases} 1, & \text{if } \alpha = 0 \\ \lambda_\beta(k) + \frac{\sum_\beta \alpha}{\alpha}, & \text{otherwise} \end{cases}, \text{ where } \alpha_\beta = \overline{\left| \nabla_\theta \hat{\mathcal{L}}_\beta(k) \right|}$$

    **end if**

    Update the parameters $\theta$ via gradient descent, e.g., $\theta(k+1) = \theta(k) - \eta \nabla_\theta \mathcal{L}(k)$.

**end for**

**Return:** $\theta$

---

Algorithm 1 exhibits DC-PINNs characteristics that set it apart from conventional learning methods. First, the computation points $x_{\{f,h\}}$ for the derivatives of the MLP do not correspond with the points of the training dataset $x_0$. The algorithm adjusts the derivatives to fit wide mesh grids in the defined space, thereby capturing derivative data across a wide array of input features. Secondly, the objective function $\mathcal{L}$ does not depend only on the MLP's direct output but also on its derivatives as specified in Eq.equation 9, all of which depend on identical network parameters. DC-PINNs facilitate balancing among categorized losses in addition to enhanced individual losses, which consist of PDE residuals and various scaled losses resulting from the violation of inequality constraints.

## 5 EXPERIMENTAL DESIGN

### 5.1 NEURAL NETWORK SETTING AND TRAINING CONFIGURATION

In the experiments, the network architecture $\varphi$ is a deep setting with four hidden layers ($L = 5$), each containing 32 neurons and a hyperbolic tangent activation function for smooth activations, following Wang et al. (2023) for specific measures to improve learning efficiency and accuracy in selecting appropriate architectures. We also employ Glorot initialization for network parameters and the Adam optimization (Kingma & Ba, 2014) with a weight decay setting, which starts with a learning rate $\eta = 10^{-3}$ and an exponential decay with a decay rate of 0.9 for every 1000 decay steps. The hyperparameters used are $p_m, p_\lambda = 100$, and $k_{\max} = 10000$. The compared models in the result section are MLP $\mathcal{L} := \mathcal{L}_0 + \mathcal{L}_b$, PINNs $\mathcal{L} := \mathcal{L}_0 + \mathcal{L}_b + \mathcal{L}_f$, and DC-PINNs $\mathcal{L} := \lambda_0 \hat{\mathcal{L}}_0 + \lambda_b \hat{\mathcal{L}}_b + \lambda_f \hat{\mathcal{L}}_f + \lambda_h \hat{\mathcal{L}}_h$. Training data is prepared using equally distributed points for initial and boundary conditions ($N_0 = N_b = 101$) and containing square mesh grids for PDE residuals and inequality constraints, i.e., $N_f = N_h = 101 \times 101$. To evaluate approximation ability, the evaluation errors between predictions and answers are calculated using mesh grids as the same grids of constraints.

In computing, differentiable operators have been developed in JAX/Flax Bradbury et al. (2018); Heek et al. (2023), which can efficiently calculate exact derivatives using automatic differentiation. The experiments are conducted using Google Colab[1], which offers GPU computing on the NVIDIA Tesla T4 with a video random access memory of 15 GB. The complete codebase for this study is available at [GitHub].

# 6 NUMERICAL EXPERIMENTS

To demonstrate the effectiveness of the proposed framework, we conduct a series of numerical experiments on several benchmark problems related to quantitative finance: heat diffusion, local volatility surface calibration, and incompressible flow dynamics with complex geometries and nonlinear constraints. We compare our approach to existing PINN-based approaches, including derivative profile comparisons on a function of trained networks.

## 6.1 ONE-DIMENSIONAL HEAT EQUATION IN THERMODYNAMICS

The heat equation is a classic example of a parabolic partial PDE problem Cannon (1984). It is a suitable and well-established problem for illustrating the robustness of PINNs. Consider an infinitesimally thin steel beam heated at its centre by a heat source. The heat at the centre will spread over the steel beam while the edges are kept at zero temperature, ensuring that the temperature reaches zero at an infinite final time. The problem setup is as follows:

$$f(x,t) = \frac{\partial u}{\partial t} - \lambda \frac{\partial^2 u}{\partial x^2}, \tag{14}$$

$$\text{s.t.} \quad u(x,0) = \sin(\pi x), \ u(0,t) = u(1,t) = 0,$$
$$\frac{\partial^2 u}{\partial x^2} \leq 0, \frac{\partial u}{\partial t} \leq 0, \tag{15}$$

where $x, t \in [0,1]$. The solution to this heat equation is given by $u(t,x) = e^{-\lambda \pi^2 t} \sin \pi x$. We can add derivative constraints, as shown in the second line of equation 15, based on the fact that the specified derivatives of the analytical solution should be negative in the defined space. This is also intuitively convincing because the high heat at the centre gradually spreads to the edges, maintaining solutions as parabolic curves over the domain throughout the entire timespan with heat reduction. We set coefficient $\lambda = 0.1$ in this experiment.

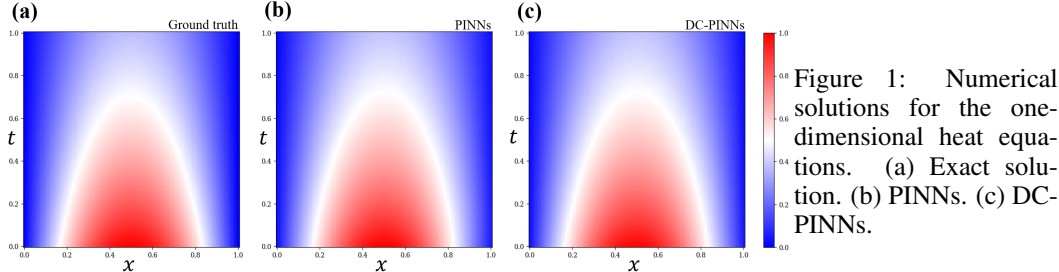

Figure 1: Numerical solutions for the one-dimensional heat equations. (a) Exact solution. (b) PINNs. (c) DC-PINNs.

Figure 1 compares the temperature fields learned by the proposed DC-PINNs framework with those of the standard PINNs approach. Both methods appear to achieve sufficient fitting and do not exhibit significant differences in accuracy when fitting the overall temperature profile.

To further investigate the impact of incorporating inequality constraints in DC-PINNs, Figure 2 illustrates the derivative profiles of the learned temperature fields with respect to the spatial coordinate $x$ at different time snapshots. These sensitivity profiles highlight the key differences between the PINNs and DC-PINNs solutions. The standard PINNs generate temperature profiles with distinct regions of non-physical positive in differentials $\partial^2 u/\partial x^2$ and $\partial u/\partial t$. In contrast, DC-PINNs consistently produce sensitivity profiles that adhere to non-positivity constraints in differentials $\partial^2 u/\partial x^2$

---

[1]Google Colab. http://colab.research.google.com

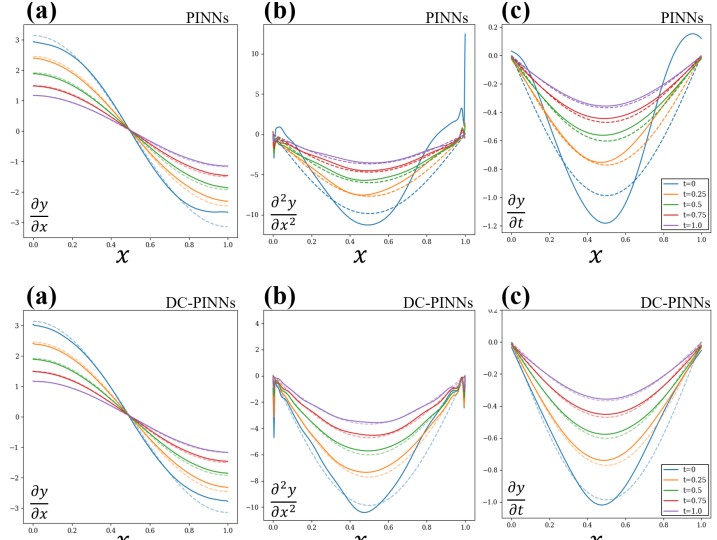

Figure 2: Derivative profiles of the learned temperature fields at different time snapshots for PINNs (upper row) and DC-PINNs (bottom row), aligned with exact solutions (dashed lines). (a) First-order derivatives with respect to $x$. (b) Second-order derivatives with respect to $x$. (c) First-order derivatives with respect to $t$.

and $\partial u/\partial t$ across all time snapshots while closely matching the ground truth profiles. This demonstrates the effectiveness of the DC-PINNs framework in enforcing inequality constraints during the learning process, resulting in physically consistent solutions. These results emphasize the importance of explicit constraint handling in physics-informed neural networks, as naively relying on the partial differential equation (PDE) residual term alone may lead to solutions that violate critical physical principles.

## 6.2 IMPLIED VOLATILITY SURFACE CALIBRATION IN FINANCE

Next, we consider the calibration problem in finance, an inverse problem to identify governing parameters in a PDE, where the option prices satisfy the PDE (i.e., the Local Volatility (LV) model introduced in Dupire et al. (1994). The implied volatility $u$ is given with respect to strike $x$ and time to maturity $t$ by,

$$f(x, t) = \frac{\partial u}{\partial t} - \frac{1}{2}\sigma_{\mathrm{LV}}^2 x^2 \frac{\partial^2 u}{\partial x^2} + rx \frac{\partial u}{\partial x}, \tag{16}$$
$$u = g(x, t, y), \quad \sigma_{\mathrm{LV}} = \phi(x, t, y, \mathcal{D}y),$$

$$\mathrm{s.t.} \quad u_{t=0} = (s_{t_m} - x)^+, \; u_{x=0} = s_t,$$
$$\frac{\partial u}{\partial x} \leq 0, \; \frac{\partial^2 u}{\partial x^2} \geq 0, \; \frac{\partial u}{\partial t} \geq 0, \tag{17}$$

where $u$ represents the option price calculated by the function $g(\cdot)$, which is plugged into the formula in equation 21. The conversion function $\phi(\cdot)$ is with respect to $y$ using equation 24. Assuming European (call) options with various strikes and time to maturity $x, t \in [0, 1]$, $s_0 = 0.1$, $t_m = 1$, $r = 0.1$, and $s_t = s_0 e^{rt}$. Synthetic data is prepared as option premiums ($u$) using the SABR model (Hagan et al., 2002). It is noted that this problem is different from previous examples since the feasible solution by optimizations affects the designed parameters of PDEs. Figure 3 illustrates

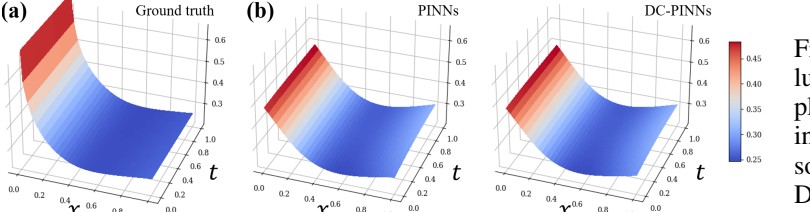

Figure 3: Numerical solutions ($\sigma_{\mathrm{LV}}$) for the implied surface calibration in finance. (a) Analytical solution. (b) PINNs. (d) DC-PINNs.

the numerical solutions obtained by each method. Similar to the previous examples, both methods

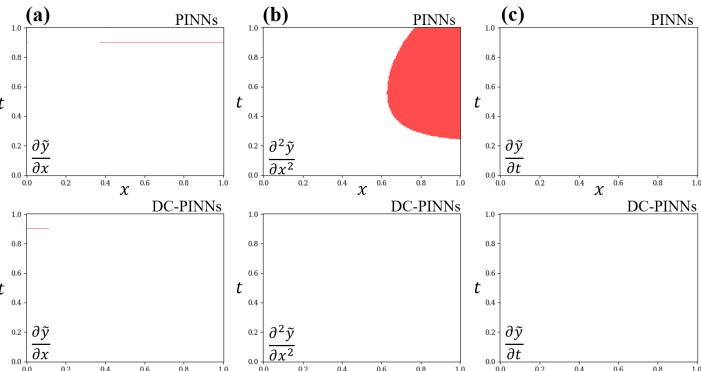

Figure 4: Heatmaps of the inequality derivatives conditions of the learned local volatilities for PINNs (upper row) and DC-PINNs (bottom row), the area that violates constraints is coloured red. (a) First-order derivatives with respect to $x$. (b) Second-order derivatives with respect to $x$. (c) First-order derivatives with respect to $t$.

in the calibration problem also appear to achieve appropriate fitting and do not exhibit significant differences in accuracy when predicting the overall parameter profiles.

As with other experiments, we compare the performance of DC-PINNs that incorporate nonlinear constraints by investigating the derivative profiles of the obtained surface. Figure 4 illustrates the area of heatmaps of the inequality derivatives conditions of the learned local volatilities as same as previous examples. The results demonstrate that the proposed DC-PINNs framework captures the solution while satisfying the nonlinear constraints, outperforming the traditional PINNs approach. Furthermore, the DC-PINNs framework demonstrates versatility, as it can be applied to calibration problems of design parameters in PDEs, including data-driven solutions.

## 6.3 NAVIER-STOKES EQUATIONS IN FLUID DYNAMICS

To demonstrate the applicability of DC-PINNs to more complex physical systems, this study considers the incompressible Navier-Stokes equations, which are fundamental yet challenging in fluid dynamics. The flow past a 2D circular cylinder is modeled, representing the well-known phenomenon of von Kármán vortex street. The governing equations in convective form are given by:

$$\begin{cases} f^u = \frac{\partial u}{\partial t} + \mu_1 \left( u\frac{\partial u}{\partial x} + v\frac{\partial u}{\partial y} \right) + \frac{\partial p}{\partial x} - \mu_2 \left( \frac{\partial^2 u}{\partial x^2} + \frac{\partial^2 u}{\partial y^2} \right) = 0 \\ f^v = \frac{\partial v}{\partial t} + \mu_1 \left( u\frac{\partial v}{\partial x} + v\frac{\partial v}{\partial y} \right) + \frac{\partial p}{\partial y} - \mu_2 \left( \frac{\partial^2 v}{\partial x^2} + \frac{\partial^2 v}{\partial y^2} \right) = 0 \end{cases} \tag{18}$$

$$\text{s.t.} \quad (u,v)|_{x=-15} = (u,v)|_{y=-8,8} = (1,0), \ \left.\frac{\partial u}{\partial x}, \frac{\partial v}{\partial x}\right|_{x=25} = p|_{x=-15} = 0,$$

$$u = v = 0 \text{ on the cylinder surface}, \tag{19}$$

$$\nabla \cdot \mathbf{V} = \frac{\partial u}{\partial x} + \frac{\partial v}{\partial y} = 0, \quad \left|\frac{\partial u}{\partial x}\right|, \left|\frac{\partial v}{\partial y}\right| \leq 1,$$

where $\mathbf{V} = (u,v)$ is the velocity field, $p$ is the pressure, and $\mu_1 = 1$ and $\mu_2 = 1/\text{Re}$ are dimensionless parameters, with Re being the Reynolds number. This problem is treated as a data-discovery problem for PDEs, where the parameters $\mu = (\mu_1, \mu_2)$ are identified from observed data. We simulate flow samples past a circular cylinder with a diameter $D = 1$ under incompressibility conditions at a Reynolds number $Re = 100$. The computational domain is a rectangle defined by $x \in [-15, 25]$ and $y \in [-8, 8]$, with the cylinder center at the origin. The training velocity field $\mathbf{V}$ is sampled on a random grid of 5,000 points within the region $x \in [1, 8]$, $y \in [-2, 2]$, and $t \in [0, 20]$ in periodic steady flow throughout. The results are compared with those obtained using a spectral/hp element method implemented in the open-source software Nektar++ (Moxey et al., 2020), as illustrated in Figure 5.

In the training, velocity and pressure fields are defined $(u, v, p) := \varphi_\theta (x, y, t)$ in DC-PINNs and $(\phi, p) := \varphi_\theta (x, y, t), (u, v) = (\partial\phi/\partial y, -\partial\phi/\partial x)$ in PINNs, following Raissi et al. (2019). In the experiment, we consider the divergence-free field and vortices shed in the wake having a characteristic size comparable to the cylinder diameter as derivative constraints, inspired by Singh & Mittal (2005), in equation 19.

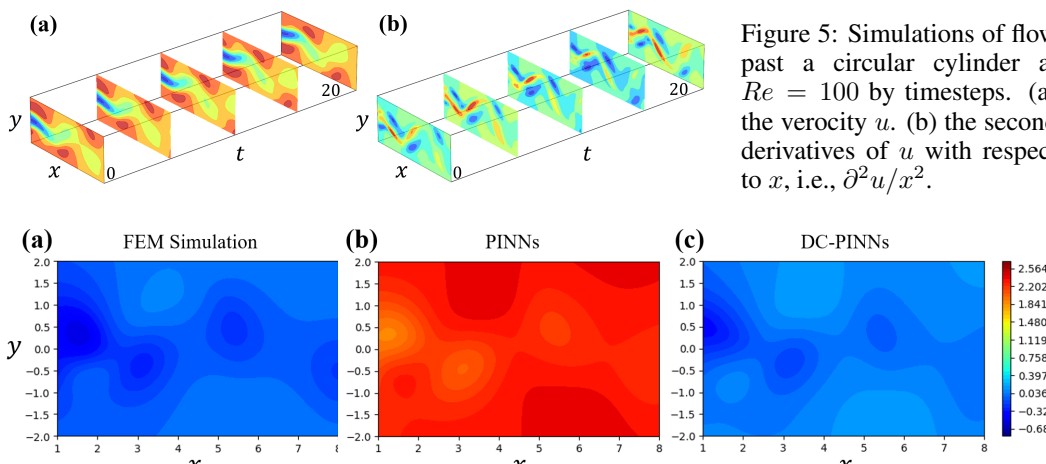

Figure 5: Simulations of flow past a circular cylinder at $Re = 100$ by timesteps. (a) the verocity $u$. (b) the second derivatives of $u$ with respect to $x$, i.e., $\partial^2 u / x^2$.

Figure 6: Comparison of trained pressure fields for flow past a circular cylinder. (a) Finite element method. (b) PINNs. (c) DC-PINNs.

Figure 6 compares the pressure fields obtained from models at a representative time step in the vortex shedding cycle. Both trained models reproduce symmetry and periodicity features. However, the scale of the numbers differs, and DC-PINNs reproduce the magnitude more accurately, indicating that DC-PINNs grasp the whole-term relationship of PDEs more precisely from the training.

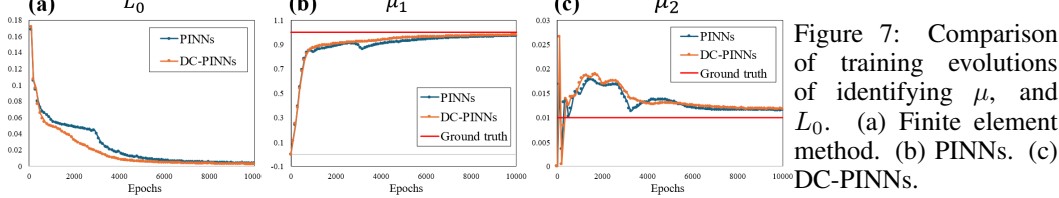

Figure 7: Comparison of training evolutions of identifying $\mu$, and $L_0$. (a) Finite element method. (b) PINNs. (c) DC-PINNs.

To evaluate the performance of DC-PINNs, Figure 7 presents the system identification learning, or data-driven discovery of PDEs, showing how the parameters $\mu = (\mu_1, \mu_2)$ describe the observed data. DC-PINNs effectively converge to the exact answers for the parameters $\mu$ during the training process. The results show that DC-PINNs can accurately capture complex flow features, including the von Kármán vortex street in the cylinder's wake. The additional physical constraints enforced by DC-PINNs help maintain solution stability and prevent non-physical artefacts that may arise in unconstrained neural network approaches.

This case study highlights the potential of DC-PINNs for solving challenging nonlinear PDEs in fluid dynamics. The framework's ability to incorporate domain-specific knowledge through additional constraints enhances the physical consistency of the solutions, making it a promising tool for various computational PDE dynamics applications.

## 6.4 COMPUTATIONAL EFFICIENCY

At last, we demonstrate the effectiveness of DC-PINNS during training and evaluate its computational efficiency. Table 1 presents the computation times required by DC-PINNs and the baseline models for calibrating the surface in 6.2 based on changes in dataset size and number of neurons. The computational efficiency of DC-PINNs is evident from their ability to handle these complex optimization challenges without significant overhead. The efficient use of automatic differentiation and the adaptive loss balancing approach contribute to DC-PINNs convergence and reduced computational overhead. DC-PINNs also exhibit reasonable scalability, as evidenced by the sublinear growth in computation time with respect to the dataset size. This scalability is crucial for handling the ever-increasing volumes of real-world data in modern scientific domains.

Table 1: The computation times (in seconds) for the training in 6.2 based on changes in dataset size (total $N$) and number of neurons.

| Models | Default | Dataset size | | ♯ of neurons | |
|---|---|---|---|---|---|
| | | Half | Quarter | Half | Quarter |
| MLP | 48.2 | 45.0 | 44.8 | 45.4 | 44.8 |
| PINNs | 95.5 | 68.4 | 56.7 | 70.6 | 57.9 |
| DC-PINNs | 122.3 | 92.6 | 80.9 | 93.2 | 82.6 |

## 7 CONCLUSION AND FUTURE WORK

In this study, we proposed an extended PINNs framework called Derivative-Constrained PINNs (DC-PINNs) to effectively solve PDEs with derivative constraints, which seamlessly integrates constraint information into the learning process through a constraint-aware loss function. The effectiveness of the DC-PINNs framework was demonstrated through a series of numerical experiments on benchmark problems related to applied physics to quantitative finance, including heat diffusion, volatility surface calibration, and the Navier-Stokes equation. The results showed that DC-PINNs outperformed standard PINNs approaches in terms of the ability to satisfy nonlinear constraints with sufficient accuracy. DC-PINNs also illustrated computational efficiency in handling optimization without significant overhead by the use of automatic differentiation and adaptive loss balancing techniques.

The study emphasizes the importance of explicit constraint handling in PINNs, as relying solely on the PDE residual term may lead to solutions that violate critical physical principles. The DC-PINNs framework opens up new possibilities for solving constrained PDEs in multi-objective optimization problems. However, further investigation is needed to assess the scalability of the framework to high-dimensional and more complex constraint types. Future work could explore efficient sampling strategies and adaptive collocation point selection to mitigate the curse of dimensionality and improve accuracy and computational efficiency.

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

# APPENDIX

## A   IMPLIED VOLATILITY SURFACE CALIBRATION IN FINANCE

### A.1   PROBLEM SETTING

The implied volatility surface is a model of values resulting from European option prices in quantitative finance. We are given a complete filtered probability space $\left(\Omega, \mathcal{F}, (\mathcal{F}_t)_{t \in [0,T]}, \mathbb{P}\right)$ and that $\mathbb{P}$ is an associated risk-neutral measure. The price of a European call option $C$ at time t is defined as

$$C = e^{-r\tau}\mathbb{E}\left[(S_T - K)^+ \mid \mathcal{F}_t\right],\tag{20}$$

where $S_t$ is the underlying price at $t$, $K$ is the strike price, $\tau = T - t$ is the time to maturity $T$, and $r$ is the risk-free rate. In Black & Scholes (1973), the implied volatility $\sigma_{\text{imp}}$ leads to the modelled price with $K, \tau \in [0, \infty)$ as the Black-Scholes (BS) formula,

$$C_{\text{BS}}\left(\sigma_{\text{imp}}\right) = S_t N\left(d_+\right) - e^{-r\tau} K N\left(d_-\right),$$
$$d_\pm = \frac{\ln(e^{-r\tau} S_t/K) \pm \left(\sigma_{\text{imp}}^2/2\right)\tau}{\sigma_{\text{imp}}\sqrt{\tau}},\tag{21}$$

where $N(\cdot)$ is the cumulative normal distribution function.

To expand the generalization for $K$ and $\tau$, Dupire et al. (1994) proposed the Local Volatility (LV) model, in which the European option prices satisfy the PDE,

$$rK\frac{\partial C}{\partial K} - \frac{1}{2}\sigma_{\text{LV}}^2(K,\tau)K^2\frac{\partial^2 C}{\partial K^2} + \frac{\partial C}{\partial \tau} = 0, \tag{22}$$

with initial and boundary conditions given by

$$C_{\tau=0} = (S_T - K)^+, \quad \lim_{K\to\infty} C = 0, \quad \lim_{K\to 0} C = S_t. \tag{23}$$

Plugging it into the formula in equation 21, one obtains a conversion function $\sigma_{\text{imp}}$ and $\sigma_{\text{LV}}$ with respect to $K$ and $\tau$,

$$\sigma_{\text{LV}}^2(K,\tau) = \frac{\sigma_{\text{imp}}^2 + 2\sigma_{\text{imp}}\tau\left(\frac{\partial\sigma_{\text{imp}}}{\partial\tau} + rK\frac{\partial\sigma_{\text{imp}}}{\partial K}\right)}{1 + 2d_+K\sqrt{\tau}\frac{\partial\sigma_{\text{imp}}}{\partial K} + K^2\tau\left(d_+d_-\left(\frac{\partial\sigma_{\text{imp}}}{\partial K}\right)^2 + \sigma_{\text{imp}}\frac{\partial^2\sigma_{\text{imp}}}{\partial K^2}\right)} \tag{24}$$

to be fit with the PDE equation 22.

We can define the calibration as identifying the multivariate function respecting the prices $\Phi(x)$, associated with implied volatility surface as a function $\varphi(x) \geq 0$ with inputs $x := (K, \tau)$,

$$\Phi(x) = C_{\text{BS}}(K, \tau, \varphi(x)). \tag{25}$$

The inverse problem of the implied volatility surface is that, given limited options prices, we would like to identify the implied volatility function with respect to $x$, redefined as $\sigma_{\text{imp}}(x)$, to fit with that premium resulted by $C_{\text{BS}}$ also satisfy PDE. Based on (21$\sim$24), once $\varphi$ is determined, we can analytically obtain the option price $\Phi$. Furthermore, $\Phi$ is second differentiable whenever $\varphi$ is second differentiable, allowing the representation of the PDE in equation 22.

## A.2 No-Arbitrage Constraints for European Options

The option prices should obey the constraints imposed by no-arbitrage conditions, which are essential financial principles posit that market prices prevent guaranteed returns above the risk-free rate. This study considers the necessary and sufficient conditions for no-arbitrage presented in Carr & Madan (2005). This allows us to express the call option price as a two-dimensional surface appropriately. The necessary and sufficient conditions for no-arbitrage are represented as non-strict inequalities for several first and second derivatives,

$$-e^{-r\tau} \leq \frac{\partial C}{\partial K} \leq 0, \quad \frac{\partial^2 C}{\partial K^2} \geq 0, \quad \frac{\partial C}{\partial \tau} \geq 0. \tag{26}$$

From the above, no-arbitrage conditions require these derivatives to have a specific sign. The standard architecture does not automatically satisfy these conditions when calibrating with a loss function simply based on the mean squared error (MSE) for the prices.

## A.3 The SABR model

The SABR model in Hagan et al. (2002) is a typical parametric model, which can capture the market volatility smile and skewness and reasonably depict market structure. When $F_t$ is defined as the forward price of an underlying asset at time $t$, the SABR model is described as

$$dF_t = \alpha_t F_t^\beta dW_t^1, \quad d\alpha_t = \nu\alpha_t\, dW_t^2, \tag{27}$$
$$\langle dW_t^1, dW_t^2 \rangle = \rho dt.$$

Here, $W_t^1$, $W_t^2$ are standard Wiener processes, $\alpha_t$ is the model volatility, $\rho$ is the correlation between the two processes, and $\nu$ is analogous to vol of vol. The additional parameter $\beta$ describes the slope of the skewness. Essentially, the IV in the SABR model is given by a series expansion technique associated with volatility form of Black (1976)

$$\sigma(K,\tau) = \frac{\alpha\left(1 + \left(\frac{(1-\beta)^2}{24}\frac{\alpha^2}{(FK)^{1-\beta}} + \frac{1}{4}\frac{\rho\beta v\alpha}{(FK)^{(1-\beta)/2}} + \frac{2-3\rho^2}{24}v^2\right)\tau\right)}{(FK)^{(1-\beta)/2}\left[1 + \frac{(1-\beta)^2}{24}\ln^2\frac{F}{K} + \frac{(1-\beta)^4}{1920}\ln^4\frac{F}{K}\right]}\frac{z}{\chi(z)}, \tag{28}$$

$$z = \frac{v}{\alpha}(FK)^{(1-\beta)/2}\ln\frac{F}{K}, \quad \chi(z) = \ln\left(\frac{\sqrt{1-2\rho z + z^2} + z - \rho}{1-\rho}\right).$$

### A.4 BLACK-SCHOLES MODEL IN QUANTITATIVE FINANCE

One may realise well-known parametric modelling of volatility surface, called the Black-Scholes model, as an example of solving a PDE in which inequality constraints on derivatives play a major role; we consider the European (call) option pricing problem, which is a typical quantitative finance problem governed by the Black-Scholes model Black & Scholes (1973). The Black-Scholes equation is a parabolic partial differential equation that describes the option price $u$ in terms of the price of the underlying asset $x$ and time $t$, given the following initial and boundary conditions:

$$f(x,t) = \frac{\partial u}{\partial t} + \frac{1}{2}\sigma^2 x^2 \frac{\partial^2 u}{\partial x^2} + rx\frac{\partial u}{\partial x} - ru, \tag{29}$$

$$\text{s.t.} \quad u(x, t_m) = (x_{t_m} - k)^+, \ u(0, t) = 0,$$
$$\frac{\partial u}{\partial x} \geq 0, \ \frac{\partial^2 u}{\partial x^2} \geq 0, \ \frac{\partial u}{\partial t} \leq 0, \tag{30}$$

where $k$ is the strike price, $\tau = t_m - t$ is the time to maturity $t_m$, $\sigma$ is the volatility, and $r$ is the risk-free rate. For the purpose of this study, we consider a specific type of European option with $x, t \in [0, 1]$, $k = 0.5$, $t_m = 1$, $r = 0.1$, and $\sigma = 0.3$. The inequalities in the second line of Equation equation 30 represent the necessary and sufficient conditions for no-arbitrage, which ensure that the option price is consistent with other financial strategies. These conditions are fundamental principles that the price of the financial product must satisfy. The no-arbitrage constraints are expressed as inequalities for the first and second derivatives with respect to the underlying asset $x$ and time $t$, as discussed in Section A.2. The exact solutions for the option price with respect to $x$ and $t$ are provided by the Black-Scholes formula Black & Scholes (1973),

$$u(x,t) = x_t N(d_+) - e^{-r\tau} k N(d_-), \qquad d_\pm = \frac{\ln(e^{-r\tau} x_t/k) \pm (\sigma^2/2)\tau}{\sigma\sqrt{\tau}}, \tag{31}$$

where $N(\cdot)$ is the cumulative normal distribution function.

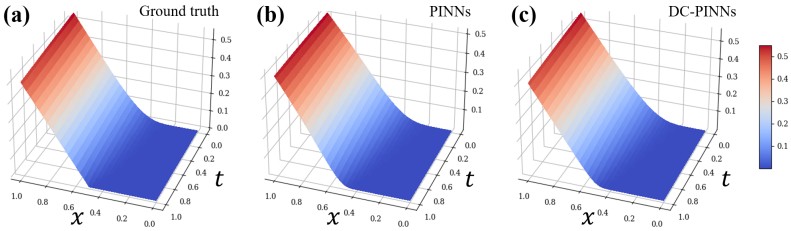

Figure 8: Numerical solutions for the Black-Scholes model in finance with derivative-constrained conditions. (a) Exact solution. (b) PINNs solution. (c) DC-PINNs solution.

Figure 8 illustrates the numerical solutions obtained by each method in a defined space. Both methods in the Black-Scholes pricing problem also appear to achieve appropriate fitting and do not exhibit significant differences in accuracy when fitting the overall price profile.

To further investigate the impact of incorporating inequality constraints in DC-PINNs as in the previous experiment, as in the previous example, Figure 9 illustrates heatmaps of the inequality derivatives conditions of the trained models; the area that violates constraints are coloured red. These sensitivity profiles also highlight the differences in solutions, which is that DC-PINNs consistently produce sensitivity profiles that adhere to the no-arbitrage constraints across almost time snapshots, although PINNs violate the inequality condition of derivatives on the wide area.

## B INCOMPRESSIBLE FLOW DYNAMICS IN PHYSICS

### B.1 NUMERICAL SETUP AND DATA GENERATION

**Computational Domain** The computational domain for our Navier-Stokes simulation is a rectangular region defined by $x \in [-15, 25]$ and $y \in [-8, 8]$. A circular cylinder with a diameter $D = 1$ is positioned at the origin $(0, 0)$. This setup allows for the observation of the von Kármán vortex street phenomenon in the cylinder's wake as stated in García (2020).

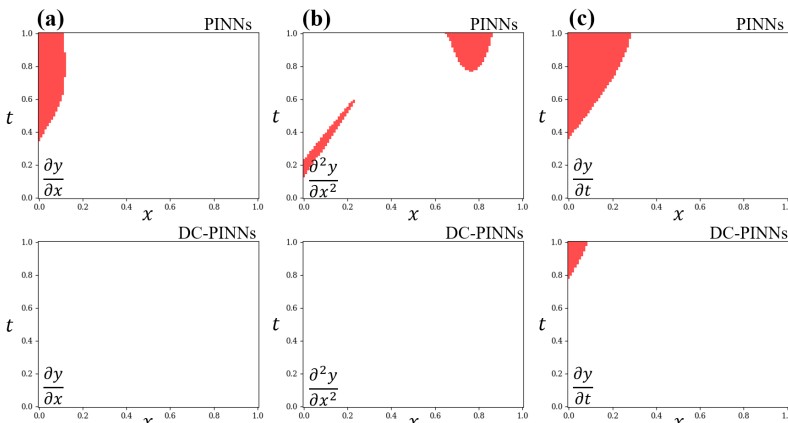

Figure 9: Heatmaps of the inequality derivatives conditions of the Black-Scholes option pricing for PINNs (upper row) and DC-PINNs (bottom row), where the area that violates constraints is coloured red. (a) First-order derivatives with respect to $x$. (b) Second-order derivatives with respect to $x$. (c) First-order derivatives with respect to $t$.

**Nektar++ Simulation Details**  We use the spectral/hp element method implemented in the open-source software Nektar++ (Moxey et al., 2020) to generate high-fidelity simulation data. The following details describe our numerical setup:

- Mesh Generation: The solution domain is discretized in space by a tessellation consisting of triangular elements. This mesh is designed to capture the flow features accurately, with refinement near the cylinder and in the wake region.
- Spatial Discretization: Within each triangular element, the solution is approximated as a linear combination of a hierarchical, semi-orthogonal Jacobi polynomial expansion. This high-order approximation allows for an accurate representation of the flow field with relatively few elements.
- Boundary Conditions: We apply the following boundary conditions (BCs): 1. No-slip conditions on the cylinder surface: homogeneous Dirichlet BC $V = 0$. 2. Far-field conditions at the top and bottom boundaries: Dirichlet BC $(u, v) = (1, 0)$. 3. Inflow condition at the left boundary: Dirichlet BC $(u, v) = (1, 0)$. 4. Outflow condition at the right boundary: static pressure $p = 0$.
- Reynolds Number: The simulation is conducted at a Reynolds number $Re = 100$, which is known to produce a stable von Kármán vortex street.
- Time Integration: We use a second-order implicit-explicit time-stepping scheme. The time step is chosen to ensure stability and accuracy, typically $\Delta t = 0.025$.
- Convergence Criteria: The simulation is run until a statistically steady state is reached, typically for about 800 time units after the simulation starts.
- Gradients generation: Each partial differential number based on Navier-Stokes PDEs is calculated using the numerical gradient function in the FieldConvert modules.

**Data Extraction and Preprocessing**  Following the Nektar++ simulation, we extract training data for the DC-PINNs using a targeted approach. We sample the velocity field $\mathbf{V} = (u, v)$ on a random grid of 5,000 points as observed samples and calculation grids for PDEs residuals and errors of derivative-constraints within $x \in [1, 8]$ and $y \in [-2, 2]$, capturing the near-wake flow behaviour. Data snapshots are collected at $0.1$ time unit intervals to represent temporal evolution.

B.2  DC-PINN IMPLEMENTATION

Our DC-PINN architecture utilizes a fully connected neural network with 3 input neurons $(x, y, t)$, 8 hidden layers of 20 neurons each, and 3 output neurons $(u, v, p)$ to compare PINNs implementation by Raissi et al. (2019). We employ Tanh activation functions and implement the architecture using JAX for automatic differentiation in PDE residuals and constraints. The total loss function is defined as:

$$L_{\text{total}} = \lambda_0 \hat{\mathcal{L}}_0 + \lambda_{f^u} \hat{\mathcal{L}}_{f^u} + \lambda_{f^v} \hat{\mathcal{L}}_{f^v} + \lambda_{h^{\text{div}}} \hat{\mathcal{L}}_{h^{\text{div}}} + \lambda_{h_x^{\text{vort}}} \hat{\mathcal{L}}_{h_x^{\text{vort}}} + \lambda_{h_y^{\text{vort}}} \hat{\mathcal{L}}_{h_y^{\text{vort}}} \tag{32}$$

Here, $\hat{\mathcal{L}}_0$ measures prediction-simulation mismatch, $\hat{\mathcal{L}}_{f^u}, \hat{\mathcal{L}}_{f^v}$ represents Navier-Stokes equation residuals, $\hat{\mathcal{L}}_{h^{\mathrm{div}}}$ enforces the divergence-free constraint, and $\hat{\mathcal{L}}_{h_x^{\mathrm{vort}}}, \hat{\mathcal{L}}_{h_y^{\mathrm{vort}}}$ implements the vorticity constraint. The $\lambda_i$ coefficients are balancing hyperparameters. We utilize the Adam optimizer (Kingma & Ba, 2014) with a learning rate of $10^{-3}$ and cosine annealing, full batch training, and $10{,}000$ epochs. Each iteration involves sampling domain points, computing forward passes, calculating losses, and updating parameters via backpropagation.

Physical constraints are implemented using automatic differentiation. The divergence-free constraint penalizes non-zero $\partial u/\partial x + \partial v/\partial y$, while the vorticity constraint penalizes $\partial u/\partial x$, $\partial y/\partial y$ exceeding flow-characteristic thresholds.

## B.3 COMPARATIVE ANALYSIS

We evaluate DC-PINNs against standard PINNs and the finite element method (FEM). Our analysis encompasses accuracy through RMSE errors in velocity and pressure fields, computational efficiency via training and inference time comparisons, and constraint satisfaction by assessing adherence to the divergence-free condition and other physical constraints. This comprehensive comparison elucidates the relative strengths of each method in solving the Navier-Stokes equations. The following shows slice of the fields including all the derivatives after quantitative comparison.

| RMSE | Training | | Validation | |
|---|---|---|---|---|
| | PINNs | DC-PINNs | PINNs | DC-PINNs |
| $u$ | **0.0376** | 0.0416 | **0.0376** | 0.0413 |
| $v$ | 0.0445 | **0.0436** | **0.0439** | 0.0448 |
| $p$ | 2.3618 | **0.1299** | 2.3619 | **0.1293** |
| $u_x$ | **0.0805** | 0.0867 | **0.0376** | 0.0877 |
| $v_x$ | 0.1345 | **0.1341** | 0.1362 | **0.1300** |
| $p_x$ | 0.0550 | **0.0530** | 0.0562 | **0.0525** |
| $u_y$ | **0.1513** | 0.1621 | **0.0376** | 0.1642 |
| $v_y$ | **0.0809** | 0.0854 | **0.0824** | 0.0863 |
| $p_y$ | **0.0521** | 0.0537 | **0.0529** | 0.0535 |
| $u_t$ | 0.6082 | **0.5877** | **0.5908** | 0.5912 |
| $v_t$ | 0.3926 | **0.3777** | 0.3837 | **0.3834** |
| $u_{xx}$ | **0.2689** | 0.2825 | **0.2718** | 0.2908 |
| $v_{xx}$ | 0.5158 | **0.3528** | 0.5129 | **0.4798** |
| $u_{yy}$ | **0.9340** | 0.9413 | **0.8870** | 0.9288 |
| $v_{yy}$ | **0.3271** | 0.3536 | **0.3378** | 0.3543 |
| $f_u$ | 0.6194 | **0.5924** | 0.6002 | **0.5595** |
| $f_v$ | 0.4247 | **0.4054** | 0.4141 | **0.4119** |
| Training / Inference Time (s) | 735.96 | **225.13** | 3.8963 | **0.9833** |
| Divergence-free Error | **0.0000** | 0.0253 | **0.0000** | 0.0250 |
| Vortices shed Error (sum.) | 0.0000 | 0.0000 | 0.0000 | 0.0000 |

Table 2: Quantitative comparison of methods. **Bold** value shows lower (better) value of RMSE.

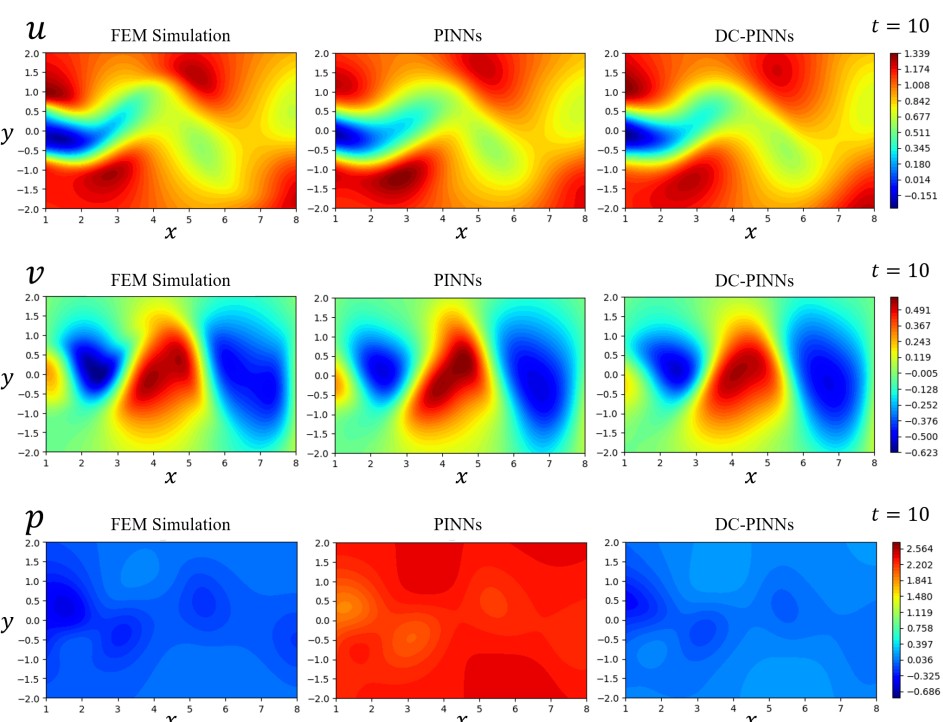

Figure 10: Comparison of FEM Simulations, standard PINNs, and DC-PINNs (1/6).

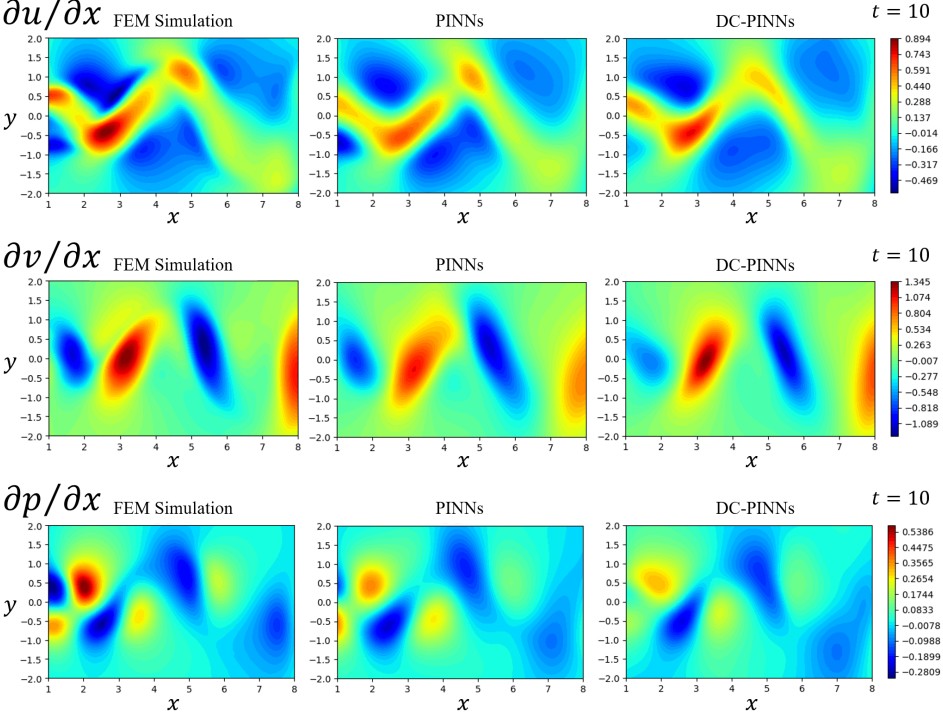

Figure 11: Comparison of FEM Simulations, standard PINNs, and DC-PINNs (2/6).

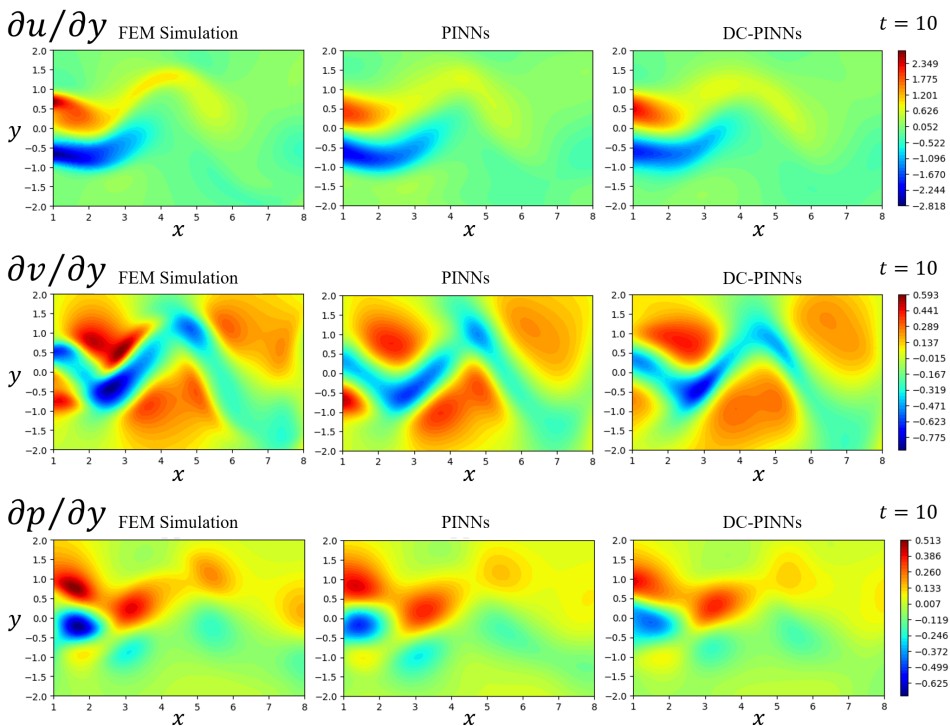

Figure 12: Comparison of FEM Simulations, standard PINNs, and DC-PINNs (3/6).

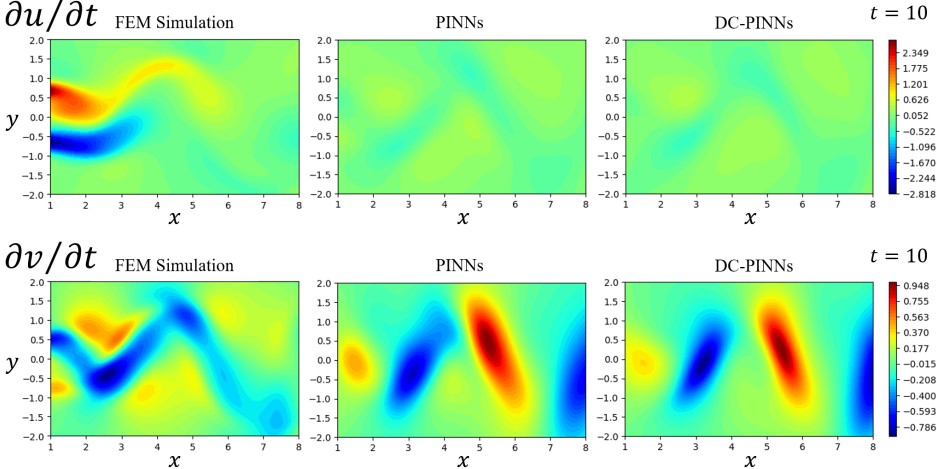

Figure 13: Comparison of FEM Simulations, standard PINNs, and DC-PINNs (4/6).

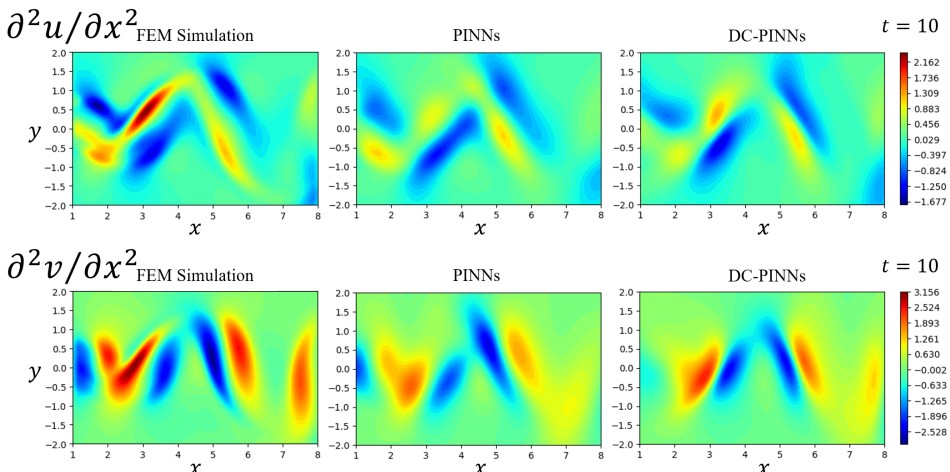

Figure 14: Comparison of FEM Simulations, standard PINNs, and DC-PINNs (5/6).

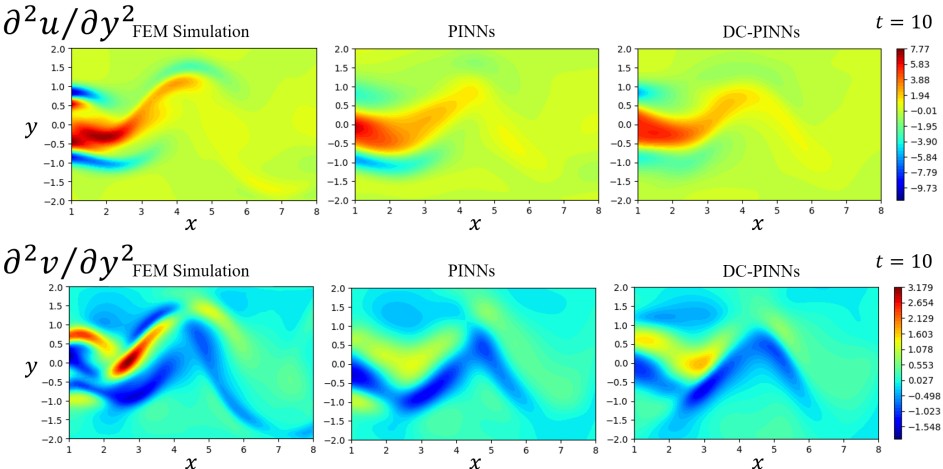

Figure 15: Comparison of FEM Simulations, standard PINNs, and DC-PINNs (5/6).