# OpenReview forum: "DC-PINNs: Physics-Informed Neural Networks for Solving Derivative-Constrained PDEs"
_ICLR.cc/2025/Conference — ICLR 2025 Conference Withdrawn Submission_

### Official Review · Reviewer_eYDa · 2024-10-29

**Soundness:** 3
**Presentation:** 4
**Contribution:** 2
**Rating:** 3
**Confidence:** 4

**Summary:**

The paper extends PINNs to derivative constrained PDEs by introducing a term in the PDE loss. Specifically, the authors target solving PDE with inequalities on the derivative constraining the solution. The proposed method is then validated on several examples.

**Strengths:**

The paper is well-written and easy-to-follow. The contribution is simple to understand and to my knowledge, no work have studied thus kind of PDE solving problem with derivatives constraints.
Several examples are shown to illustrate the method.
The example chosen to evaluate are relevant and taken from several literature.
The method is well-described and experiment settings are detailed.

**Weaknesses:**

The contribution consists in the and to adding a term in the PINNs loss to model the addition of the constraints as already made in the standard PINNs proposed by Raissi et al. These constraints can be linked to Boundary condition problems, which is a well-known context of PINNs. In such context, the derivative constraints are treated the same as in the proposed method (initial and boundary residuals). The loss balancing algorithm are taken from existing works.
In the experiments, if other solution can be found by PINNs by not considering the added constraints, doesn’t it mean that the problem is ill-posed? Meaning that several solutions exist for the given PDE. This is unfair in my opinion to compare the proposed algorithm with method which (i) do not have any information about the added constraint (ii) have to solve ill-posed problems.
Doesn’t any baseline exist to solve this type of PDE-constrained problems that could be added as a comparison?

**Questions:**

- Are samples used during training? If so, where are they located? How many of them are used?
- Why aren’t MLP results displayed and commented?
- How does the different models behave when adding too much constraints? i.e. when the system does not admit any solution?
- In the example 6.3, why the settings for PINNs and DC-PINNs are not the same? (last paragraph, velocity and pressure fields) What are the results when aligning settings?
- In example 6.3, It seems that PINNs successfully learns the parameters of the PDEs? Could the author elaborate on the relative performances of PINNs wrt DC-PINNs? What are the advantages of DC-PINNs in this context?
- In example 6.4, it seems that DC-PINNs is much longer to train. Could you elaborate about this aspect? Why does the training time increases this much?

Additional remarks:
-	Typo line 249 with “Eq.equation 9”

**Details Of Ethics Concerns:**

No reproducibility or ethic statement provided in the paper

---

### Official Review · Reviewer_FmP8 · 2024-10-29

**Soundness:** 2
**Presentation:** 3
**Contribution:** 2
**Rating:** 3
**Confidence:** 4

**Summary:**

This paper develops a Derivative-Constrained PINNs framework to incorporate additional inequality constraints, beyond the typical PDE, boundary, and initial condition constraints, into the optimization process of PINNs. The primary contribution is the introduction of a self-adaptive loss balancing technique that automatically adjusts the relative weighting of various loss penalty terms, as well as the individual weighting within each type of penalty term. The authors demonstrate the effectiveness of their proposed method through several practical problems, highlighting DC-PINNs' ability to recover PDE solutions while satisfying specified inequality conditions on derivatives.

**Strengths:**

The paper is well-organized and clearly written, making it easy to follow overall. It provides strong motivation for incorporating inequality derivative constraints into the optimization process. The proposed method is clearly explained and accessible, with practical and well-known benchmark problems that demonstrate its effectiveness in addressing real-world challenges.

**Weaknesses:**

•	Novelty: Inequality-constrained PINNs have been studied in existing work, such as Lu et al. [1]. Although Lu et al. [1] addresses inverse problems, while this work directly solves a PDE, similar problems are considered using different methods. Additionally, the benchmark method from Lu et al. [1] should also be included for comparison.

•	Insufficient Experimentation: For example, in the heat equation experiment, the heat equation coefficient $\lambda$ is set to 0.1. However, PINNs have well-known failure modes: when PDE coefficients are small, PINNs can recover the PDE solution, but they fail when PDE coefficients are large, as noted by Krishnapriyan et al. [2]. The proposed method should demonstrate robustness across a range of coefficient values. Additional, more robust experiments to hyperparameters such as NN size would be good.

[1] Lu, Lu, et al. "Physics-informed neural networks with hard constraints for inverse design." SIAM Journal on Scientific Computing 43.6 (2021): B1105-B1132.

[2] Krishnapriyan, Aditi, et al. "Characterizing possible failure modes in physics-informed neural networks." Advances in neural information processing systems 34 (2021): 26548-26560.

**Questions:**

In the traditional penalty method, the penalty coefficient is required to go to infinity for the transformed unconstrained problem to converge to the original constrained problem. Here, the method you are using is still a penalty method. Nevertheless, the penalty coefficients $\lambda_{\beta}$ or $m_{\beta}^{i}$ in your work may not keep increasing if $\nabla \hat L_{\beta} (k)$ is small, even if the constraint violation remains large. I wonder if this could occur due to a complex loss landscape or potential ill-conditioning issues such as when the PDE coefficient is large.

---

### Official Review · Reviewer_3M6x · 2024-11-02

**Soundness:** 2
**Presentation:** 3
**Contribution:** 1
**Rating:** 3
**Confidence:** 5

**Summary:**

This paper considers the problem of incorporating derivative constraints when solving PDEs using PINNs. It incorporates those constraints as additional terms into the loss function and proposes to use loss balancing techniques between not only loss terms, but also the samples within each loss. The proposed method is compared to normal PINNs for different PDEs.

**Strengths:**

- The proposed method is better at satisfying derivative constraints than regular PINNs (that do not incorporate any derivative information).
- The paper proposed loss balancing methods appears to have beneficial performance effects (although testing is severely limited).
- The structure and writing is mostly clear (despite the presence of several typos in the formal parts).

**Weaknesses:**

The major weaknesses of the paper are: (1) incremental contribution; (2) limited motivation; (3) scarce experimental validation; (4) weak guarantees and formalism.

1. **Incremental contribution:** The method used by the work to address derivative constraints is no different than that from regular PINNs in (Raissi et al., 2019), namely, combining automatic differentiation and penalties. The penalty used is in fact typical from classical constrained optimization methods, such as augmented Lagrangian (see, e.g., Bertsekas, “Constrained Optimization and Lagrange Multiplier Methods,” 1982). In other words, no additional contribution is needed to account for such constraints.
Additionally, the proposed balancing method is essentially a combination of (Wang et al. 2023) and (McClenny & Braga-Neto 2020). In particular, the sample balancing adaptation (12) is the same as in (McClenny & Braga-Neto 2020), while the loss balancing uses the method proposed in (Wang et al. 2023), replacing a moving average and l2 norm by an exact average and l1 norm. Neither change is clearly motivated (see (4) below). While the combination of these two methods is novel, it is unclear why or when it is needed (see (3) below). While the manuscript acknowledges its inspiration in (Wang et al. 2023) and (McClenny & Braga-Neto 2020), it does not properly state these similarities (see (2) below). For instance, the idea of balancing losses and samples within losses already appeared, e.g., in (Wang et al. 2023).

2. **Limited motivation:** the paper does not properly motivate the need for derivative constraints, either in its introduction or in the experiments. This is a particularly critical issue in view of point (1) above. For instance, the derivative constraint proposed for the heat equation appears to arise from the knowledge of the phenomenon, i.e., of the PDE solution. This is in fact stated in the text: “We can add derivative constraints [...] based on the fact that the specified derivatives of the analytical solution should be negative in the defined space.” In other cases, the derivative constraints appear to be part of the boundary conditions of the problem, e.g., in the finance and Navier-Stokes case. Comparison to regular PINNs in all these settings is therefore completely unfair (see (3) below), seen as regular PINNs either do not use the additional information (heat equation) or does not even tackle the same problem (finance/Navier-Stokes). What is more, the no-arbitrage constraints in the finance application do not appear to be of the same type considered throughout the text, since they must consider a two-sided inequality, namely (26). In the case of the Black-Scholes model, the motivation for using NNs is unclear since a closed-form solution can be obtained somewhat straightforwardly. Finally, the paper provides very limited discussion on its relation to prior work, particularly when it comes to other approaches to incorporate constraints when solving PDEs. There are a plethora of papers on incorporating boundary conditions and other constraints, such as Chalapathi et al., ICLR’24; Gao et al., J. Comp. Phys.’21; Psichogios et al., AIChE J.’92; Lu et al., 2021; Basir et al., J. Comp. Phys.’22, to name only a few. Any of these would also provide much better baselines (see (3) below).

3. **Scarce experimental validation:** The only comparison to other methods in the paper is with traditional PINNs. No other methods for incorporating constraints into PINNs are considered. Regular PINNs are, however, a poor baseline since it does not even tackle the proposed PDE problems. This is particularly critical in the finance and Navier-Stokes applications, where the PDE problems considered have boundary conditions incorporating derivative inequalities (see (16-17) and (18-19)). In that sense, it is not surprising that, e.g., DC-PINNs perform better than PINNs in Figure 2 or Figure 6, since PINNs do not account for the derivative conditions in the problem formulations. Additionally, there is no experimental validation of the proposed weight balancing scheme. This is an issue particularly since it is not well-motivated (see points (1) and (4)). A better baseline would be, e.g., using PINNs with the derivative penalty term with fixed weights. This would also help understand the role and necessity of the different weighting schemes and better motivate their combination. Since no theoretical motivation or guarantee is provided (see (4) below), this lack of experimental evidence is all the more critical.

4. **Weak guarantees and formalism:** While the paper is for the most part well-written, its lack of formalism makes it hard to follow at times. Firstly, several typos throughout the derivations make the presentation confusing, e.g., the use of $u$ for the PDE solution in line 78 immediately followed by calling it $y$ in (1-2) and throughout the rest of the text. Taken formally, (7-8) also do not make sense since the “inequality” to be satisfied in (8) is $x \geq 0$ and the argument of $\gamma$ in (7) is always non-negative, namely, $| h(\phi(x)) |^2$. I assume this is a typo since it is corrected in (11). What is more, the derivative constraints considered in the work are of the form $h(\phi(x)) \geq 0$. Hence, the loss $\mathcal{L}_h$ should have a negative sign (seen as $\mathcal{L}$ is being minimized and the value of $h$ must be increased in order to satisfy the constraint). These recurring typos in the formalism of the paper not only confuse the reader, but also suggest that the manuscript needs more attention before being published.
Another critical point is the motivation for the use of the l1 norm (absolute value) in (13) as opposed to the l2 norm (squared values) from Wang et al., 2023. The text states that “the main reason for the choice of absolute values [...] is to avoid overlooking outlier violations of the inequality constraints.” This justification does not make sense to me. In fact, the squared value gives considerably more weight to outliers than the absolute value, which is the whole reason why the latter is used in robust statistics, e.g., the Huber loss, or for using the l1 norm in compressed sensing. The manuscript goes on to justify that “most elements of $\nabla\mathcal{L}_h$ are assumed to be zero values in almost all cases.” They would therefore not need the l1 geometry to force their sparsity, in which case the squared value would once again make sense (as is the case in, e.g., elastic net).
Finally, it should be noted that the use of penalties with weight adaptation heuristic does not enforce the derivative constraints, but only penalizes their value. This can be seen in Figure 4, where even DC-PINNs do not fully satisfy the constraints. What is more, it could be that satisfying the derivative constraints would lead to violating the PDE or boundary conditions (this can be done by, e.g., making $\lambda_h \to \infty$). It is hard to tell from Figure 3 whether that is the case since no numerical values are provided for the errors. In fact, neither PINNs nor DC-PINNs appear to fit the ground truth well in this case. In that sense, it would also be more informative if Figure 4 showcased the value of the constraints so as to show how much it is violated, i.e., having a continuous rather than binary heatmap. This would be in line with the rationale provided in the text on line 143 stating that “setting large loss weights can cause an ill-conditioned problem. On the other hand, when small loss weights are chosen, the estimated solution may violate the inequalities.” As it is, this statement is not justified, either theoretically or empirically. In particular, without any theoretical guarantee or substantial experimental validation (see (3) above) of the proposed methods, it is hard to justify the “derivative-constrained” name.

**Questions:**

See Weaknesses. Here are additional minor points:

1. How are the different terms in the PINN baseline weighted? This is quite standard and known to have big impacts on performance. How does the DC-PINN compare to PINN including the derivative penalty with a fixed weight chosen by, e.g., cross-validation?
1. How is the constraint (26) handled in the finance application? In other words, how is the two-sided inequality handled when the penalty in (7) is designed for $D(y(x)) \geq 0$?
1. On line 453, it says “Figure 6 compares the pressure fields obtained from models at a representative time …”. What is meant by a representative time step? It would be good to have some performance metric aggregated for all times.
1. In the caption of Figure 5, should “verocity” read “velocity”?
1. In general, the labels and legends of the figures are too small (especially the legends).
1. The bibliography should be updated to reflect the current status of papers. E.g., (McClenny & Braga-Neto 2020) has already been published for over a year.

In view of these issues, I do not recomment that the paper be published in its current state.

---

### Official Review · Reviewer_1jUb · 2024-11-04

**Soundness:** 2
**Presentation:** 2
**Contribution:** 2
**Rating:** 3
**Confidence:** 4

**Summary:**

This paper introduces a variation on PINN that adds derivative constraints to ensure accuracy of MLP approximation of the solution to a given PDE. This variation introduces the ReLU of the constraint residuals as the additional loss term together with the three vanilla PINN losses. To balance these loss terms, this paper also proposes weights for all four losses, and an adaptive algorithm to adjust these weights during training. The new approach is tested on three types of PDE: heat, Black-Scholes and 2D Navier-Stokes.

**Strengths:**

The proposed method for balancing the load is presented in a logical way and seems  to improve the previous balancing algorithms by Wang, et al.

The method is implemented on three sets of benchmark problems.

**Weaknesses:**

(1) It is unclear how realistic the problem set up is. For a genuine PDE solver with initial and boundary data, derivative information within the domain is typically unknown. Incorporating this information into the loss function introduces data leakage, as it relies on knowledge from output quantity of interest. Here is a detailed argument about the meaning of having this extra information.

Take the 1D heat equation in the paper as an example, if we change the initial data to $ u(t=0, x) =\sin(2\pi x) $.
The derivative constraints $ \frac{\partial^2 u}{\partial x^2} \leq 0, \quad \frac{\partial u}{\partial t} \leq 0 $
will destroy the existence of the unconstrained solution $u(t,x)=e^{-4\pi^2\lambda t}\sin(2\pi x)$ for all $t\geq0$, but the unconstrained problem itself is a well-posed PDE problem and vanilla PINN solves it at acceptable accuracy.

If constraints are adjusted accordingly to allow existence of solution, it becomes a clear source of data leakage (setting correct derivative bounds to allow existence of solution).

Some of the PDEs have solutions intrinsically large derivatives, so that adding derivative constraints modifies the genuine solution.
We have an example from original PINN paper:
$
    u_t + u u_x - \frac{0.01}{\pi} u_{xx} =0 \qquad (t,x)\in[0,1]\times[-1,1]
$, $u(0, x) = - \sin(\pi x),$ and $u(t, -1) = u(t, 1) = 0.$
Set the constraints are posed as
$ \left| \frac{\partial u}{\partial x} \right| \leq \text{bound}, $. Then,
even if we notice that setting $\text{bound}<\pi$ leads to the initial condition violating the constraints so that the bound is adjusted to be larger without prior knowledge of the solution, $\frac{\partial u}{\partial x}$ will eventually reach the bound as a shockwave, so the solution from DC-PINN beyond this period will diverge from the true (unconstrained) solution or/and violate the derivative constraints.

Due to the possible data leakage problem above, improved accuracy may solely source from this extra info, diminishing the effectiveness of the loss balancing mechanism as another claimed contribution of the paper

(2) Presentation quality
- Equation (1) and (2) have an undefined variable $y$. From context, it is expected to be $u$.

- Equation (14) has $u(x, t)$ as the argument order ($x$ first, $t$ second) but in the description (line 298), it appears as $u(t, x)$, which are inconsistent. It is recommend to be consistent.

- Figure 7 captions are wrong: none of figures mention FEM, but caption says (a) is FEM. From context, it is expected to be (a)$L_0$, (b)$\mu_1$, (c) $\mu_2$.

(3) Need more comparisons:

The paper mentions that the proposed load balancing algorithm is derived from the method of Wang et al 2023. It is expected to have some comparisons with the previous method.

Moreover, there are other baseline methods that can be compared with the proposed load balancing technique:

- Kim, et al, DPM: A novel training method for physics-informed neural networks in extrapolation, AAAI, 2021.

- Yao, et al, Multiadam: Parameter-wise scale-invariant optimizer for multiscale training of physics-informed neural networks, ICML, 2023.

**Questions:**

The main question is included in the first point of the weaknesses above.

---

### Note · Authors · 2024-11-25

**Comment:**

After careful consideration of the reviewers' feedback, we have decided to withdraw our submission from the conference. We sincerely appreciate the detailed reviews and constructive suggestions provided by the reviewers. Their insights have highlighted important areas for improvement that will help us strengthen the theoretical foundations and experimental validation of our work. We look forward to addressing these points comprehensively in future submissions.

We would like to express our gratitude to the program committee and reviewers for their time and effort in providing valuable feedback.

**Withdrawal Confirmation:**

I have read and agree with the venue's withdrawal policy on behalf of myself and my co-authors.